# GROUP-BASED INTERLEAVED PIPELINE PARALLELISM FOR LARGE-SCALE DNN TRAINING

**Pengcheng Yang, Xiaoming Zhang, Wenpeng Zhang, Ming Yang, Hong Wei**
Ant Group, China
`yangpc615@gmail.com, xiaominglan.zhang@antgroup.com`
`zhangwenpeng0@gmail.com, vincent.ym@antgroup.com`
`weihong9646@hotmail.com`

## ABSTRACT

The recent trend of using large-scale deep neural networks (DNN) to boost performance has propelled the development of the parallel pipelining technique for efficient DNN training, which has resulted in the development of several prominent pipelines such as GPipe, PipeDream, and PipeDream-`2BW`. However, the current leading pipeline PipeDream-`2BW` still suffers from two major drawbacks, i.e., the excessive memory redundancy and the delayed weight updates across all stages. In this work, we propose a novel pipeline named WPipe, which achieves better memory efficiency and fresher weight updates. WPipe uses a novel pipelining scheme that divides model partitions into two groups. It moves the forward pass of the next period of weight updates to the front of the backward pass of the current period of weight updates in the first group, retains the order in the second group, and updates each group alternatively. This scheme can eliminate half of the delayed gradients and memory redundancy compared to PipeDream-`2BW`. The experiments, which train large BERT language models, show that compared to PipeDream-`2BW`, WPipe achieves $1.4\times$ acceleration and reduces the memory footprint by 36%, without nearly sacrificing any final model accuracy.

## 1 INTRODUCTION

Several recent lines of research (Liu et al., 2019; Yang et al., 2019; Lan et al., 2019; Raffel et al., 2019; Brown et al., 2020; Lin et al., 2021) on various application domains have collectively demonstrated that larger DNN models can yield better performance. This creates an emerging trend in scaling up the number of model parameters, resulting in very large DNNs, the memory footprint of which will be beyond the limit of a single accelerator.

To resolve the aforementioned problem, researchers and practitioners have focused on model parallelism, which allows to further scale up model parameters significantly by partitioning a large model over multiple accelerators/workers. However, the conventional model parallelism, which includes inter-layer model parallelism (Narayanan et al., 2021) and intra-layer model parallelism (Shoeybi et al., 2019), either suffers from low resource utilization or high communication overhead. Recently, some studies proposed the pipeline-parallel technique to accelerate conventional model-parallel training. The pipelining technique can effectively improve the utilization of computing resources by scheduling different workers to consume different mini-batches in parallel. However, naive pipelining cannot be directly used for DNN training due to two problems: (1) staleness of weight updates (Harlap et al., 2018), which is caused by the inconsistency of the weight versions used in a forward pass and its corresponding backward pass, and (2) excessive in-flight activations, which are heaped up by continuous forward activations waiting for their corresponding backward passes in pipelining.

To address the staleness problem, some workable pipelines have been proposed with different emphases on low memory footprint and high throughput. GPipe proposes the periodic pipeline flushing technique, which maintains only one weight version. This technique can achieve a low memory footprint and almost the same weight update semantics as data parallelism, but at the cost of introducing bubble overhead (Huang et al., 2019), thus limiting throughput. PipeDream proposes the weights stashing (Harlap et al., 2018) technique to eliminate bubble overhead, resulting in higher

throughput; however, the multiple weight versions incur a high memory footprint. To further reduce the weight versions maintained in PipeDream, PipeDream-2BW was proposed in which the double-buffered weight update technique (Narayanan et al., 2021) is used, which largely reduces the number of weight versions and enables similar weight update semantics as that in data parallelism.

To reduce in-flight activations, GPipe and PipeDream-2BW adopt activation recomputation (Chen et al., 2016; Jain et al., 2019). Recomputation enables the forward pass to execute without retaining the internal activations, which are recomputed in the corresponding backward pass. This method of being recomputed when needed can effectively alleviate the stacking up of in-flight activations and greatly reduce the memory footprint. Although both GPipe and PipeDream-2BW utilize the recomputation technique, PipeDream-2BW has fewer in-flight activations than GPipe owing to the more timely execution of the backward pass in its pipeline. Regarding Pipedream, unfortunately, it doesn't take effective measures on this issue, which prevents it from training a larger model.

Currently, PipeDream-2BW is the pipeline-parallel system with the best overall performance. However, it still suffers from two main problems: delayed weight updates and excessive memory redundancy on weight versions and activations (Narayanan et al., 2021). In this paper, we propose a novel pipeline, named WPipe, to solve these two problems. In WPipe, model partitions are divided into two groups, namely $G_0$ and $G_1$, which share the same workers. In the execution pipeline, the forward pass of the next update period is moved to the front of the backward pass of the current update period in $G_0$, and the execution order in $G_1$ is retained, and then, $G_0$ and $G_1$ are updated alternatively. We named the alternate update technique for the grouping weights as double-grouped weight updates (2GW). Compared to 2BW, 2GW has two main improvements: (1) 2GW realizes the same weight update semantics as that in data parallelism and only maintains one weight version without introducing bubble overhead in $G_1$. (2) 2GW reduces half of the in-flight activations since only half of the weights are involved in the training at a time.

Additionally, we incorporate some conventional but effective communication optimization techniques including the overlap of computation and communication, hybrid parallelism, and heterogeneous communication to further reduce communication overhead. Finally, our throughput experiments show that WPipe achieves acceleration of $7.1\times$ compared to that of PipeDream and $2.1\times$ compared to that of GPipe for Bert192 with 1.4 billion parameters. When training models with up to 5.5 billion parameters, which cannot be trained by GPipe and PipeDream, WPipe is $1.4\times$ faster than PipeDream-2BW. Memory experiments show that compared to PipeDream-2BW, WPipe reduces the memory footprint by 35.8% for Bert96. Convergence experiments show that WPipe can achieve similar final accuracy to data parallelism.

## 2 BACKGROUND

In this section, we briefly introduce the related techniques of model parallelism.

**Model parallelism.** Model parallelism (Chen et al., 2018; Chilimbi et al., 2014; Dean et al., 2012; Jia et al., 2018) partitions a large model into multiple parts and assigns them to different workers. Each worker is responsible for their own weight updates and sending and receiving intermediate activations and gradients. The conventional model parallelism includes inter-layer model parallelism and intra-layer model parallelism (Shoeybi et al., 2019). The former suffers from underutilized resources as only one worker is active at a time, as shown in Figure 1a. The latter requires all-to-all aggregation with intermediate activations/gradients for each layer. The all-to-all communication overhead, which grows in proportion to the number of layers, makes it difficult to expand to deeper layers, especially in heterogeneous network interconnections.

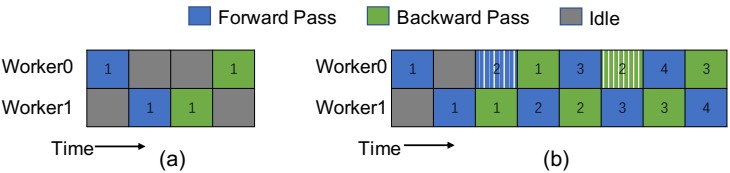

Figure 1: The execution timelines of inter-layer model parallelism and naive pipeline parallelism.

**Pipeline Parallelism.** Pipeline parallelism can effectively improve the utilization of computing resources of inter-layer model parallelism by scheduling different workers to consume different mini-batches simultaneously. However, naive pipeline parallelism suffers from a staleness issue for weight updates. As shown in Figure 1b, when the backward pass of the latter mini-batch is executed, its corresponding weights could be updated by the prior mini-batch. For example, for mini-batch 2, the weights used in the backward pass were updated by mini-batch 1.

Existing solutions tackle the issue by trading off throughput and memory footprint. GPipe proposes the periodic pipeline flushing technique (Huang et al., 2019). As shown in Figure 2a, it splits a mini-batch into multiple smaller micro-batches, which are injected into the model continuously to achieve pipelining. Within each pipeline period, GPipe can accumulate gradients across micro-batches and flush synchronously. In this way, it achieves the same weight update semantics as data parallelism and only maintains one weight version. However, in GPipe, bubble overhead (Huang et al., 2019) limits throughput, especially when a large mini-batch cannot be supported. PipeDream-flush is essentially the same as GPipe (Narayanan et al., 2021), and its main improvement is to move forward the backward pass, thereby releasing in-flight activations as soon as possible and reducing memory footprint.

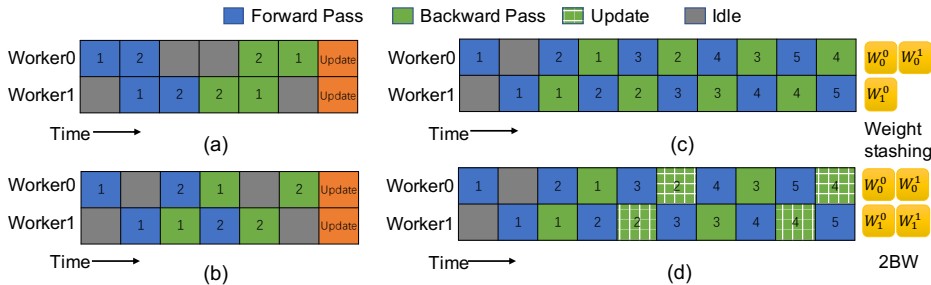

Figure 2: Timelines of various pipeline-parallel executions: (a) GPipe updates the weights in a mini-batch. (b) PipeDream-flush moves backward pass forward compared to GPipe. (c) PipeDream implements weight stashing to update the weight immediately by default. (d) PipeDream-2BW realizes periodic updates through gradient accumulation. For example, when the update period is 2, micro-batches 2 and 4 update weights in the figure.

PipeDream uses the weight stashing technique to tackle the staleness issue without introducing the bubble overhead, and thus achieves higher throughput (Harlap et al., 2018). However, it must maintain a large number of weight versions. From Figure 2b worker $i(i = 0, 1, ...)$ comprises $N - i$ weight versions, where $N$ is the number of model partitions. This results in a high memory footprint, especially when $N$ is large. Moreover, in PipeDream, the weight update semantics has different delay terms at different stages.

PipeDream-2BW (Narayanan et al., 2021), as an upgraded version of PipeDream, has higher throughput and more memory efficiency. As shown in Figure 2c, it uses double-buffered weight updates (2BW), which is combined with gradient accumulation, to reduce effectively the number of weight versions maintained on each worker from $N - i$ to 2. Notably, in 2BW, its weight update semantics has only one weight delay term at each stage (Narayanan et al., 2021).

## 3 System Design of WPipe

In this section, we first analyze the problems of 2BW. Then, we propose double-grouped weight updates (2GW) and analyze 2GW from three aspects: model partitions grouping, effective learning, and memory footprint. Finally, we introduce several communication optimization techniques.

### 3.1 Introduction

To simplify the description, we denote the naive model-parallel timeline as $V_2$ (where "2" represents the number of model partitions), as shown in Figure 1a. Then we denote the process of cross-splicing $V_2$s to form a pipeline as $P(nV_2)$ (where $n$ represents the number of $V_2$), as shown in Figure 1b. Analyzing the execution regular of the pipeline in Figure 1b, we can find that $P(nV_2)$ can ensure that

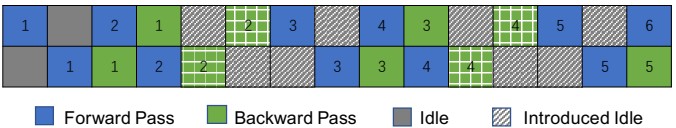

Figure 3: Separate two adjacent update periods in PipeDream-2BW by introducing idle time blocks.

the backward pass is executed immediately after the forward pass is completed so that the forward activations can be released in time to alleviate the stacking up of the activations. However, the $P(nV_2)$ cannot be used directly due to the inconsistency issue discussed above. To address the issue, 2BW uses two weight versions and combines gradient accumulation. Why does 2BW work effectively? The point is that 2BW effectively handles the intersection of adjacent update periods. As shown in Figure 2c, the forward pass for micro-batch 3 is executed before the backward pass for micro-batch 2, but its backward pass is executed after the backward pass for micro-batch 2. At this time, it is necessary to continue to use the old weight version in the period of micro-batches 3 and 4, and the new weight version (updated by micro-batch 2) will be used in the next period (micro-batches 5 and 6). Thus, if we want to further reduce the number of weight versions, we need to eliminate the intersections of micro-batches 2 and 3, 4 and 5, etc. To achieve this, we temporarily introduce idle time blocks, as shown in Figure 3 (its pipeline is the same as PipeDream-flush).

If these idle time blocks cannot be eliminated, can we fill them? Through the statistical analysis, we found that for a pipeline of PipeDream-2BW with $n$ model partitions, at least the idle time blocks shaped like $P(xV_n), x >= (n-1)$ are required to eliminate the intersections. In addition, a pipeline accumulation period has at least execution time blocks shaped like $P(xV_n), x >= n$. Obviously, when $x >= n$, they have the same shape. Thus, can we fill these idle time blocks with the pipelining of another part of the weights of the same model?

## 3.2 DOUBLE-GROUPED WEIGHT UPDATES (2GW)

In this subsection, we introduce the 2GW technique from two points: model partitions grouping and weight update semantics.

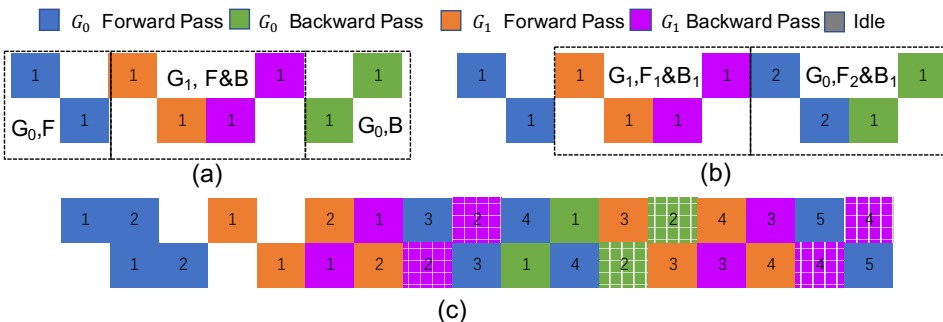

Figure 4: Derivation of WPipe pipeline: (a) Further partitioning and grouping, (b) moving the $G_0$ forward pass of the next step to the front of the backward pass of $G_0$ of the current, (c) expansion of the pipelining for (b).

**Model Partitions Grouping.** As shown in Figure 4a, based on Figure 1a, we further split each model partition into two and obtain twice the number of model partitions. Then, we divide them into two groups, $G_0$ and $G_1$, and train them on the same devices. For $G_1$, it is a typical $V_2$ structure, which can realize continuous pipeline expansion through $P$. But for $G_0$, since its forward pass and backward pass are separated, a continuous pipeline cannot be directly achieved by cross-splicing ($P$). Therefore, we first need to convert it into a $V_2$ structure. By analyzing the execution rules of the pipeline after grouping, we found that moving the forward pass of $G_0$ of the next step to the front of the backward pass of $G_0$ of the current step can form a $V_2$, as shown in Figure 4b. After doing this, we can cross-splice $G_0$ and $G_1$ respectively to obtain $P(G_0)$ and $P(G_1)$, and then continue to cross-splice their results to obtain $P(P(G_0), P(G_1))$. For $G_1$, its micro-batches 2 and 3 can be well

Table 1: Details of the memory footprint of pipeline systems of GPipe, PipeDream, PipeDream-2BW, and WPipe, where we ignore the intermediate activations for recomputation.

| PIPELINE | BUBBLE RATIO | THE SIZE OF ALL BUFFERS | ACTIVATIONS (NON-RECOMPUTATION) | ACTIVATIONS (RECOMPUTATION) |
|---|---|---|---|---|
| GPIPE | $\frac{N-1}{M+N-1}$ | 0 | $MS_a$ | $S_a$ |
| PIPEDREAM | 0 | $\frac{N+1}{2}S_m$ | $\frac{2M-N+1}{2}S_a$ | NONE |
| PIPEDREAM-2BW | 0 | $2S_m$ | $\frac{2M-N+1}{2}S_a$ | $S_a$ |
| WPIPE | 0 | $\boldsymbol{S_m}$ | $\frac{4M-N+1}{4}S_a$ | $\boldsymbol{0.5S_a}$ |

separated by the pipelining of $G_0$ ($P(G_0)$), as shown in Figure 4c. This also answers the question mentioned in section 3.1. The pipelining of $G_0$ can fill in the idle time blocks in Figure 3.

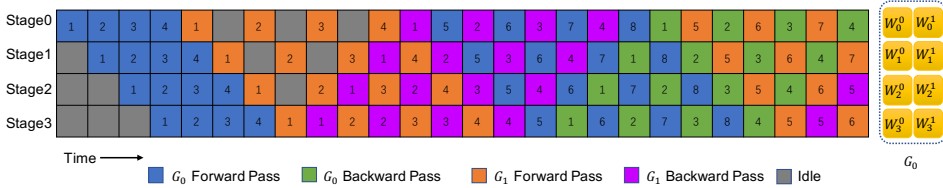

Figure 5: Timeline of execution of 2GW, where only $G_0$ needs double weight versions.

**Weight Update Semantics.** As shown in Figure 4c, $G_1$ only maintains one weight version and has the same weight update semantics as data parallelism. For $G_0$, due to the movement operation, the intersections naturally exist, such as micro-batches 3, 4, and 2 in Figure 4c. Nevertheless, $G_0$ only needs to maintain two weight versions and has only one delay term in the weight update semantics, which is the same as 2BW. If using SGD optimizer, the weight update semantics of 2GW are as follows:

$$w_{G_i}^{t+1} = w_{G_i}^t - \nu \cdot \nabla f(w_{G_i1}^{(t-1+i)}, w_{G_i2}^{(t-1+i)}, ..., w_{G_in}^{(t-1+i)}), i \in \{0, 1\}. \tag{1}$$

Other optimizers can be similarly analyzed. Figure 5 shows a more detailed pipeline-parallel timeline of 2GW. In each pipeline stage, the model partitions of $G_0$ and $G_1$ are executed alternately without introducing bubbles after the system is warmed up.

### 3.3 MEMORY FOOTPRINT ANALYSIS

The memory footprint of most pipelines is mainly divided into four parts, i.e., naive model weights, weight buffers (saving history weight versions), in-flight activations, and optimizers. Since the naive model weights and optimizers are roughly the same for each pipeline method, we mainly analyze weight buffers and in-flight activations. In WPipe, $G_1$ has no weight buffer, and the size of $G_0$'s weight buffer is 2. We use $S_p$ to represent the size of the entire model parameters, $M_{G_0}$ and $M_{G_1}$ represent the sizes of $G_0$ and $G_1$, respectively. Supposing that the model is evenly divided ($M_{G_1} = M_{G_0}$), the size of weight buffer of WPipe is $M = 2M_{G_1} = M_{G_0} + M_{G_1} = S_p$. For in-flight activations, we use $S_a$ to indicate the activations of the entire model for a micro-batch. From Figure 5, we statistically conclude that the in-flight activation amount of $stage_i$ is equal to $M * \frac{S_a}{2N}$ for $G_0$ and $(M-i) * \frac{S_a}{2N}$ for $G_1$, and of all stages is equal to:

$$\sum_{i=0}^{i=N-1} (2M-i) * \frac{S_a}{2N} = \frac{4M-N+1}{4} * S_a, \tag{2}$$

where the $N$ and $M$ indicate the number of pipeline stages and micro-batches, respectively, and $M >= N$. The same statistical method is used for GPipe, PipeDream, and PipeDream-2BW to obtain the size of all weight buffers and in-flight activations, respectively, as shown in Table 1.

**With Activation Recomputation.** From Table 1, the in-flight activations increase linearly with the number of pipeline stages, which will become a memory bottleneck when training a large model. Thus, a pipeline system must take measures to reduce in-flight activations. The activation recomputation is

an effective method widely accepted, especially for `2GW`. It can eliminate the stacking up of historical in-flight activations, leaving only the activations in an active state in the system, as shown in Table 1. From the table, we can summarize the following points: (a) compared to PipeDream-`2BW`, the weight buffer size of WPipe is reduced by half, and the superiority is more obvious compared to PipeDream. Although GPipe has no weight buffer, it has to introduce $\frac{N-1}{M+N-1}$ bubble overhead; (b) recomputation can significantly reduce in-flight activations for GPipe, PipeDream-`2BW`, and especially WPipe.

### 3.4 COMMUNICATION OPTIMIZATION

We use $C_a$ to represent the cost of intermediate activations/gradients in one communication, and then the model-parallel communication overheads of different pipeline systems are $C_{\text{WPipe}} = 2(2N_s - 1)C_a$, $C_{\text{2BW}} = 2(N_s - 1)C_a$, and $C_{\text{GPipe}} = 2(N_s - 1)C_a$, respectively, where $N_s$ indicates the number of stages. They show that with the same number of stages, the communication overhead of WPipe is twice that of GPipe and PipeDream-`2BW`. Nevertheless, in most cases, $C_a$ is small and can be ignored, especially when training some language models, such as Transformer (Vaswani et al., 2017). When $C_a$ is large, We have the following countermeasures:

**Combination with Data Parallelism.** Normally, too large $C_a$ is caused by too large a micro-batch. we can proportionally reduce the depth of the pipeline while reducing the size of the micro-batch, and then we proportionally increase the width of data parallelism to maintain the same global batch size, as shown in Figure 6. As a result, the size of the micro-batch becomes smaller, and the $C_a$ also decreases, while the number of accelerators and the global batch size remain unchanged. Of course, we need to weigh the communication overhead between model parallelism ($C_{\text{WPipe}}$) and data parallelism ($C_{\text{DP}}$) to choose the appropriate ratio of depth ($d$) and width ($w$). When $C_{\text{WPipe}}$ is greater than $C_{\text{DP}}$, we reduce the value of $d : w, d * w = N_{GPU}$. Otherwise, we increase the value of $d : w$.

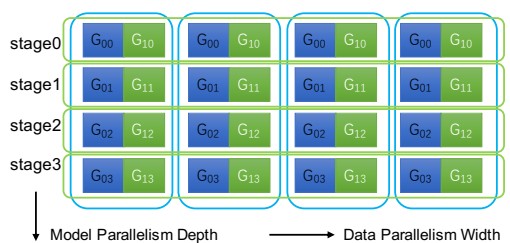

Figure 6: Layouts of model parallelism and data parallelism.

**Overlap of Computation and Communication & Heterogeneous Network Communication.** The former can almost offset the overhead of activation recomputation. The latter can effectively use heterogeneous networks to balance communication overhead. Please refer to the appendix for details. In addition, some communication compression techniques are also shown in the appendix.

## 4 EXPERIMENTS

WPipe is implemented with PyTorch-1.4 (Edward Z. Yang, 2021) and executes on two environments, i.e., a single machine with eight 16-GB V100 GPUs (Env-1) and a private cluster with $8 \times 8$V100 GPUs (Env-2).

### 4.1 QUALITY OF CONVERGENCE

In this section, we compare the convergences of WPipe and PipeDream-`2BW` by comparing the accuracy when training the same model on the same dataset with the same hyperparameters.

**Text Classification.** We finetuned BERT$_{\text{BASE}}$ (Devlin et al., 2018) and BERT$_{\text{LARGE}}$ (Devlin et al., 2018) for WPipe, PipeDream-`2BW`, and data parallelism on the QQP and MNLI tasks (Wang et al., 2018). We used respectively `bert-base-uncase` and `bert-large-uncase` pre-training weights from `transformers-3.5.0` (Wolf et al., 2020). We used Adam optimizer, a learning rate of $8 \times 10^{-5}(\nu = 8 \times 10^{-5})$ with 1000 steps warmup($ws = 1000$) and a mini-batch size of $256(b = 256)$ for BERT$_{\text{BASE}}$ and the same optimizer, $\nu = 4 \times 10^{-5}$ with $ws = 2000$ and $b = 128$ for BERT$_{\text{LARGE}}$. From Table 2, WPipe, PipeDream-`2BW`, and data parallelism have similar final accuracy. For further analysis, we continue to perform image classification experiments.

Table 2: The results of the convergence experiment of WPipe. We train the models three times with different seeds. Then, we calculated the means and standard deviations of the results. Where DP represents data parallelism and PD-2BW represents PipeDream-2BW.

| TASKS | MODEL | METRIC | DP | WPIPE | PD-2BW |
|---|---|---|---|---|---|
| QQP | BERT$_{BASE}$ | ACC | $87.66 \pm 0.06$ | $\mathbf{87.68 \pm 0.06}$ | $87.63 \pm 0.03$ |
| | | F1 | $\mathbf{83.39 \pm 0.02}$ | $83.39 \pm 0.04$ | $83.34 \pm 0.05$ |
| | BERT$_{LARGE}$ | ACC | $\mathbf{87.63 \pm 0.05}$ | $87.59 \pm 0.07$ | $87.38 \pm 0.02$ |
| | | F1 | $\mathbf{83.26 \pm 0.08}$ | $83.21 \pm 0.11$ | $82.96 \pm 0.05$ |
| MNLI | BERT$_{BASE}$ | ACC | $82.81 \pm 0.01$ | $\mathbf{83.05 \pm 0.19}$ | $82.98 \pm 0.14$ |
| | | M-ACC | $\mathbf{83.16 \pm 0.03}$ | $83.04 \pm 0.29$ | $82.82 \pm 0.26$ |
| | BERT$_{LARGE}$ | ACC | $86.16 \pm 0.12$ | $86.25 \pm 0.21$ | $\mathbf{86.29 \pm 0.12}$ |
| | | M-ACC | $86.05 \pm 0.26$ | $\mathbf{86.15 \pm 0.26}$ | $86.08 \pm 0.24$ |
| OXFORD FLOWERS102 | RESNEXT50_32X4D | ACC | $\mathbf{99.55 \pm 0.14}$ | $99.47 \pm 0.14$ | $99.51 \pm 0.12$ |
| | RESNEXT101_32X8D | ACC | $\mathbf{99.39 \pm 0.21}$ | $99.19 \pm 0.19$ | $99.19 \pm 0.19$ |
| CIFAR10 | RESNEXT50_32X4D | ACC | $97.15 \pm 0.10$ | $\mathbf{97.28 \pm 0.20}$ | $97.26 \pm 0.09$ |
| | RESNEXT101_32X8D | ACC | $98.15 \pm 0.07$ | $\mathbf{98.18 \pm 0.07}$ | $98.15 \pm 0.07$ |
| CIFAR100 | RESNEXT50_32X4D | ACC | $85.67 \pm 0.12$ | $85.62 \pm 0.09$ | $\mathbf{85.80 \pm 0.21}$ |
| | RESNEXT101_32X8D | ACC | $87.90 \pm 0.30$ | $\mathbf{88.00 \pm 0.21}$ | $87.93 \pm 0.18$ |

**Image Classification.** We finetuned, respectively, the ResNeXt50 (32x4d) (Xie et al., 2017) and ResNeXt101 (32x8d) (Xie et al., 2017) for WPipe, PipeDream-2BW, and data parallelism on the three datasets of CIFAR-10 (Krizhevsky et al., 2009), CIFAR-100 (Krizhevsky et al., 2009), and Oxford 102 Flowers (Nilsback & Zisserman, 2008). We used the pre-training weights from the torchvision (Francisco Massa, 2021), SDG optimizer, $\nu = 1 \times 10^{-2}$ with 0.05 warmup ratio and $b = 256$. From Table 2, there is still no obvious difference in their final accuracy. Thus, we continue to analyze the changes in loss of the three methods during training, while supplementing the training experiments from scratch.

**Intermediate Metrics Analysis.** As shown in Figures 7a-7c, although WPipe, PipeDream-2BW, and data parallelism have similar final loss and accuracy, the curve of WPipe is closer to data parallelism. Among them, the loss of data parallelism drops the fastest, and its accuracy rises the fastest, and WPipe is second only to data parallelism, which benefits from the fact that the weight update delay of WPipe is reduced by half, compared to PipeDream-2BW. For more loss curves and accuracy curves, please refer to the appendix.

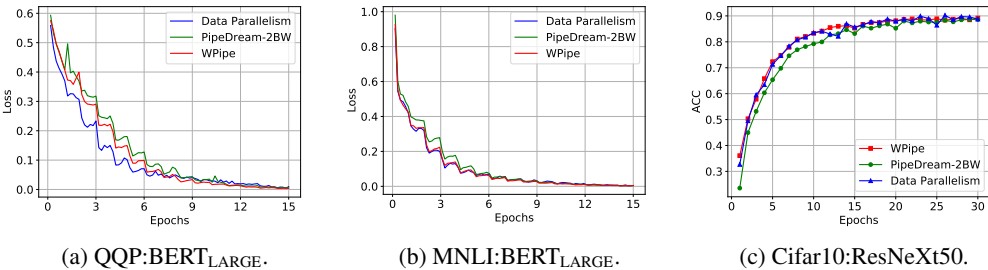

(a) QQP:BERT$_{LARGE}$.      (b) MNLI:BERT$_{LARGE}$.      (c) Cifar10:ResNeXt50.

Figure 7: Part of the training loss from Table 2 and the accuracy when training ResNeXt50 from scratch with WPipe, PipeDream-2BW, and Data Parallelism (SGD with a fixed learning rate of 0.01).

### 4.2 THROUGHPUT

In this section, we measure the throughputs of WPipe through training the large-scale DNN models on Env-1 and Env-2 and compare them with other pipeline systems. The main factors that affect throughput are batch size ($B$), the ratio of model parallelism to data parallelism ($d : w$), and the possibility to recompute ($S_a$) as presented below:

$$T = \max(T_{s \in S_a \times S_{d:w} \times S_M}); B = b * M * w, M >= d; S_a = \{True, False\};$$
$$S_{d:w} = \{H(d : w) | d * w = N_{GPU}\}; S_M = \{d, d+1, ...\}; \tag{3}$$

where $T$ is the optimal result in an optimization space $S_a \times S_{d:w} \times S_M$. $H(d:w)$ represents all isomers in the same $d:w$. $S_M$ represents the gradient accumulation period. With regards to the optimal solution of $T$, the study in (Narayanan et al., 2021) gives a workable solution, we do not do further research. Note that since $B$ directly affects the accuracy of the model, it is often a given value.

### 4.2.1 COMPARISON WITH PIPELINED APPROACHES

As shown in Figure 8, throughput experiments are carried out through training different layers of BERT and ResNeXt models on the Env-1 and Env-2. For more throughput experiment data, please refer to the appendix. In general, the communication overhead of data parallelism is larger for BERT, and model parallelism is larger for ResNeXt.

**Comparison to PipeDream-2BW.** In general, the suitable batch size is between $2^9$ and $2^{11}$. If it is too large or too small, it is not conducive to the convergence of the model. However, in most cases, when training large models, some methods cannot reach this batch size, as shown in Figure 8a. When in this range, as shown in Figure 8c, when $batch = 2^9$, WPipe is 1.4 times faster than PipeDream-2BW for Bert768 with 5.5 billion parameters. Compared to 2BW, WPipe not only reduces the cost of switching between weighted versions by half but also has a larger solution space when the batch is larger. As shown in Figure 8d, when $batch = 2^9$, WPipe can run in the configuration of $d : w = 4 : 16$ in Env-2, and 2BW due to memory limitations, the model-parallel depth is at least 8, that is, $d >= 8$, so in this case, the throughput of WPipe is higher (for convolutional networks, in general, $C_a$ is larger and $C_{DP}$ is smaller, so $d : w$ needs to be smaller to improve throughput). From Figures 8b and 8d, it can be concluded that after communication optimization, even on a convolutional network with a larger $C_a$, WPipe has a throughput not lower than PiprDream-2BW.

**Comparison to GPipe.** For GPipe, its biggest problem is that there are idle time blocks on its execution timeline. In addition, its forward activations cannot be released in time. These factors directly or indirectly affect its throughput. Compared to GPipe, from Figure 8a- 8d, WPipe is $2.1\times$ faster for Bert192 and $2.2\times$ faster for ResNeXt302.

**Comparison to PipeDream.** Due to poor memory efficiency, the larger models cannot be trained using PipeDream, such as Bert384 and Bert768. Practice shows that for ResNeXt200, ResNeXt302, ResNeXt500, and ResNeXt800, PipeDream cannot automatically split them. According to existing data, WPipe is $7.1\times$ faster for Bert192.

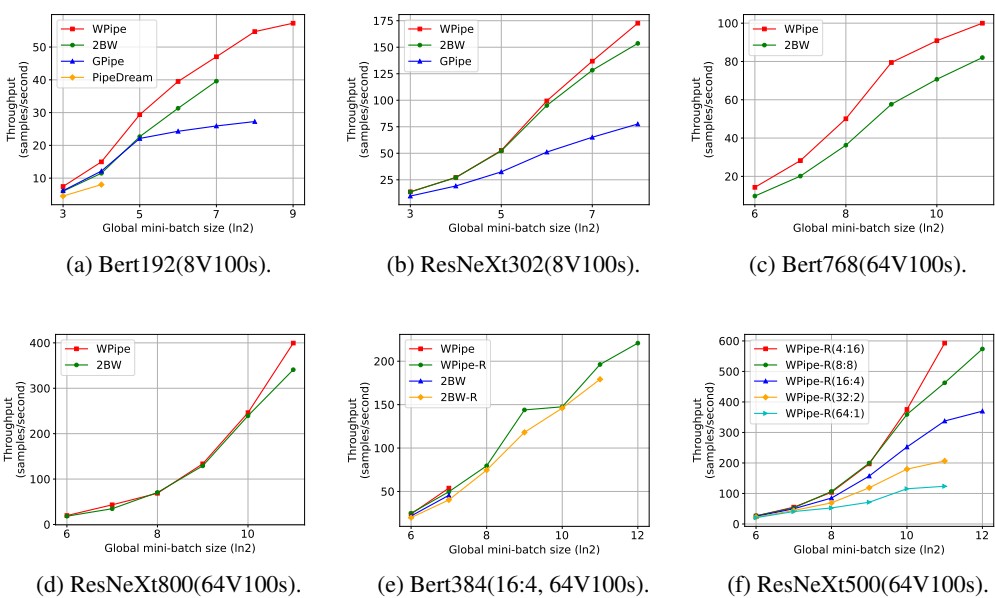

(a) Bert192(8V100s).  (b) ResNeXt302(8V100s).  (c) Bert768(64V100s).

(d) ResNeXt800(64V100s).  (e) Bert384(16:4, 64V100s).  (f) ResNeXt500(64V100s).

Figure 8: Optimal throughput for different batches in the Env-1 and Env-2. Where $S_{M:N} = \{2 : 4, 4 : 2, 8 : 1\}$ in Env-1, $S_{M:N} = \{4 : 16, 8 : 8, 16 : 4, 32 : 2, 64 : 1\}$ in Env-2. 8e-8f show the throughput changes with different configurations.

### 4.2.2 COMMUNICATION OPTIMIZATION ANALYSIS

**Recomputation.** The activation recomputation can greatly reduce memory footprint and expand mini-batch size for pipeline systems. As shown in Figure 8e, PipeDream-2BW can expand the maximum mini-batch by 16 times, and WPipe can even expand by 44 times. WPipe has a more obvious memory advantage with activation recomputation. Although the batch size is not as large as possible, usually when training large models, the batch size must be within a certain range to have better convergence. In many cases, due to memory limitations, the effective range of the batch size cannot be reached. At this time the advantage of WPipe is crucial.

**The Ratio between Model and Data parallelism.** As shown in Figure 8f, as $M : N$ decreases, the corresponding throughput increases for ResNeXt500. This happens because the communication overhead of data parallelism is smaller than that of model parallelism in the convolutional network. In this case, $M : N$ can be adjusted more widely, which is more conducive to improving throughput, e.g. $batch = 2^{11}$ in Figure 8f.

Table 3: The impact of network communication on throughput.

| Env-2(4:16) | ResNeXt500 |
| --- | --- |
| WPipe-R(A) | 607.8/3328 |
| WPipe-R(B) | 497.3/3328 |

**Heterogeneous Network Settings.** We change the communication placement of model parallelism and data parallelism in the same $M : N$ to achieve higher throughput. $A$ means that data-parallel communication takes place between machines and model-parallel communication takes place inside machines, while $B$ is the opposite. As shown in Table 3, in the case of $A$, the throughput of WPipe is higher, because the model-parallel communication is the bottleneck for ResNeXt. Compared with the inter-machine, the bandwidth inside the machine is higher and the communication speed is faster. The model-parallel communication bottleneck has been better alleviated.

### 4.3 MEMORY OPTIMIZATION ANALYSIS

In the throughput experiment, WPipe can execute on a larger batch, which also reflects the advantage of the lower memory footprint of WPipe. Here, we conduct a more detailed analysis: (1) The results of experiments show that the memory footprint of various pipeline-parallel methods is significantly reduced with activations recomputation, especially for WPipe. As shown in Figure 9, WPipe reduces the memory footprint by 65% for Bert96 on the per-GPU micro-batch of 64, and 56% for RestNeXt200 on the per-GPU micro-batch of 32. They are 47% and 29% respectively for PipeDream-2BW; (2) with the increase of batch size, the effect of recomputation on WPipe-R is more significant. From Figure 9, for Bert96, WPipe-R is reduced by -14% compared to GPipe-R on the per-GPU micro-batch of 8 (because GPipe has no weight buffer, and

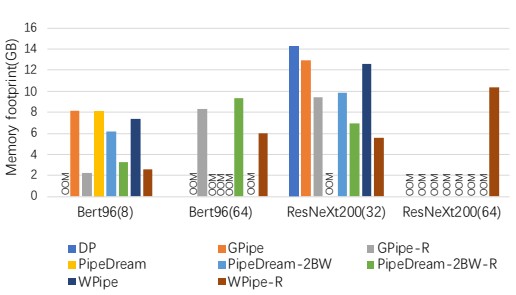

Figure 9: The Bert96 and ResNeXt200 memory footprint vary with batch size. We set $M : N$ as $8 : 1$ for Bert96 and $M : N$ as $2 : 4$ for ResNeXt200, which is the fastest configuration. We measured the average maximum memory footprint per GPU.

its initial memory footprint is lower), and on the per-GPU micro-batch of 64, it increases to 28%. Compared to PipeDream-2BW-R, under the same conditions, WPipe-R increases from 21% to 36%.

## 5 CONCLUSIONS

In this work, we proposed and implemented WPipe, a system for group-based interleaved pipeline-parallel training. Compared to the state-of-the-art approach, PipeDream-2BW, WPipe achieves better memory efficiency, higher throughput, and fresher weight updates through the double-grouped weight updates. Specifically, (1) throughput: WPipe achieves $1.4\times$ acceleration; (2) memory footprint: WPipe-R reduces the memory footprint by 36%; and (3) convergence: although WPipe and PipeDream-2BW have similar final accuracy when training, WPipe has a weight update semantics closer to data parallelism.

## ACKNOWLEDGMENTS

This work is supported by the Computational Intelligence Department of Ant Group. We thank Jiyan Jiang for his helpful discussions. We would like to thank the Aliyun EFLOPS team for their substantial support in designing the industry-leading training platform to facilitate fast trials in this work. We also thank anonymous reviewers for their insightful and valuable comments.

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

## A  APPENDIX

### A.1  OTHER EXPERIMENTAL DATA OF CONVERGENCE.

Figures 10a-10h show the training loss curves of the remaining experiment in Table 2. Figure 10i shows the F1 curve of BERT$_{BASE}$ when training from scratch. Analyzing the trend of the curve, we found that WPipe and data parallelism can converge normally, but PipeDream-2BW does not.

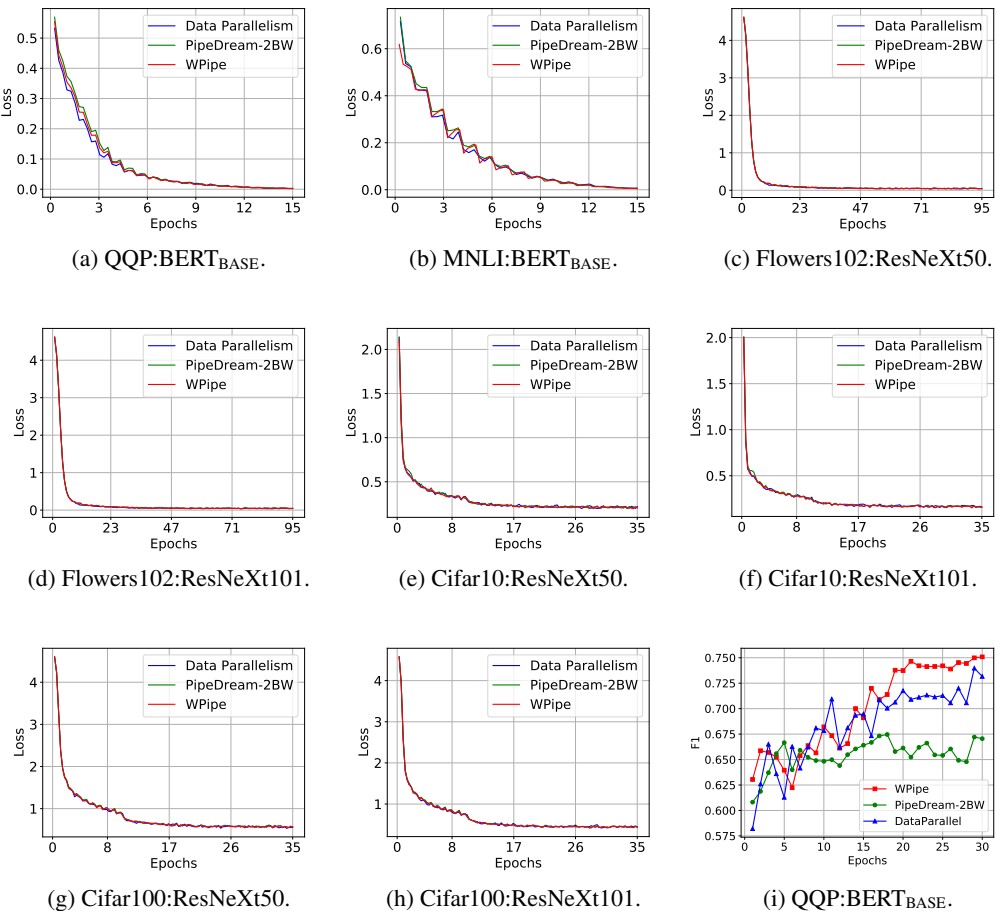

(a) QQP:BERT$_{BASE}$.  (b) MNLI:BERT$_{BASE}$.  (c) Flowers102:ResNeXt50.

(d) Flowers102:ResNeXt101.  (e) Cifar10:ResNeXt50.  (f) Cifar10:ResNeXt101.

(g) Cifar100:ResNeXt50.  (h) Cifar100:ResNeXt101.  (i) QQP:BERT$_{BASE}$.

Figure 10: The remaining part of the training loss from Table2 and the F1 when training BERT$_{BASE}$ from scratch with WPipe, PipeDream-2BW, and DataParallel (Adam with a fixed learning rate of $5 \times 10^{-6}$).

### A.2  THROUGHPUT

In the experiment, our settings of models are `hidden_size=768`, `num_attention_heads=12`, `seq_len=128` of all Bert models, and `groups=32`, `width_per_group=4` of all ResNeXt models. Regarding cluster configuration, there are 8 machines in our private cluster, and each machine has 8 GPUs with a memory size of 16G, Intel(R)Xeon(R) Platinum 8163 CPU, 512GB of RAM with a 25Gbps Ethernet interface, and 300GBps NVLink (nvl), which is 96 times the Ethernet bandwidth. In addition, the version of `PyTorch` we used was `1.4`.

As shown in Figure 11a-11d, we continue to train Bert96 and ResNeXt200 on a single machine with WPipe, PipeDream-2BW, PipeDream, and GPipe, and train Bert384 and ResNeXt500 on multiple machines with WPipe and PipeDream-2BW. The overall conclusion is the same as the above. WPipe has a more obvious acceleration effect on Transformer series networks (Bert, GDP, etc.), but has a slight acceleration on convolutional networks, compared to PipeDream-2BW.

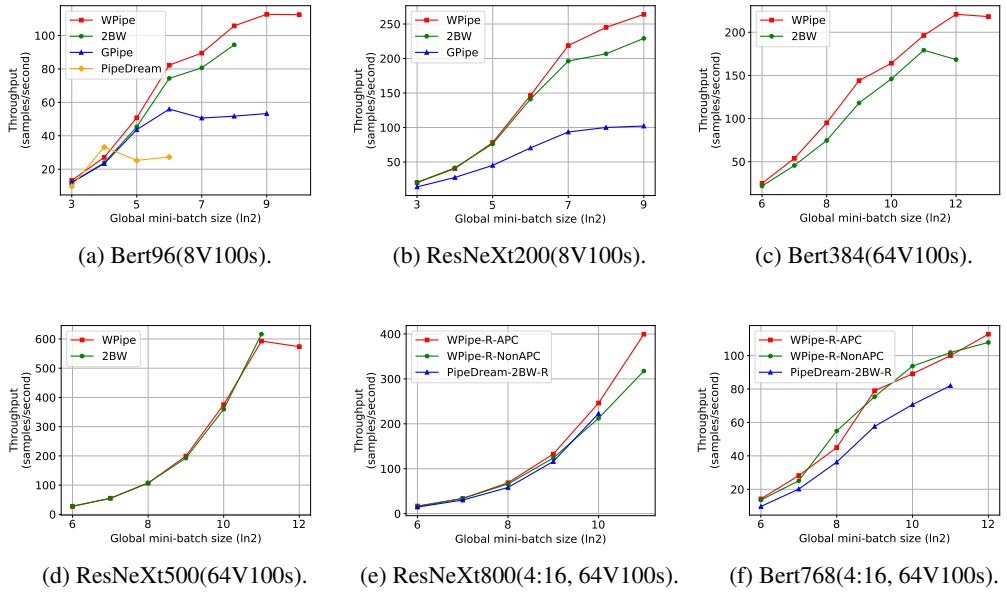

Figure 11: Optimal throughput for different batches in the Env-1 and Env-2. Where $S_{M:N} = \{2 : 4, 4 : 2, 8 : 1\}$ in Env-1, $S_{M:N} = \{4 : 16, 8 : 8, 16 : 4, 32 : 2, 64 : 1\}$ in Env-2. Figures 11e-11f show the effect of compressed model-parallel communication on different models.

## A.3   COMMUNICATION OPTIMIZATION

In this section, we use a specific example to analyze the effectiveness of the model-parallel communication compression. In addition, we add a detailed description of overlap and heterogeneous network communication.

**Communication Compression.** Referring to the automatic mixed precision algorithm of the apex (Christian Sarofeen, 2021), we implemented the automatic precision compression technique for intermediate activations/gradients, which reduced the communication overhead by nearly half across the pipeline. This offsets the communication overhead introduced by further splitting model partitions. As shown in Figure 12, the sender divides the intermediate activations/gradients by appropriate $scaler$s (powers of 2) and converts them into half-precision

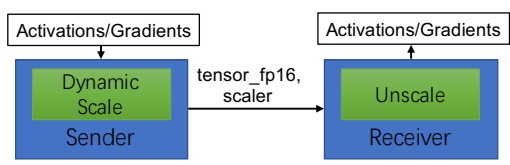

Figure 12: The communication process of the intermediate activations and gradients using automatic precision compression.

tensors. If the conversion succeeds, the sender will transmit the half-precision tensors together with the $scaler$s to the receiver. After receiving the data, the receiver restores their original precision by the $scaler$. If the conversion fails, the sender will transmit the full precision tensors.

**Automatic Precision Compression (APC).** As shown in Table 4, WPipe is $1.17\times$ faster with APC when training ResNeXt500 and the batch size equals 3328. The loss (the negative value) of accuracy, which is brought by APC, does not exceed 0.1%, as shown in Table 5. However, when the communication overhead is not large, the use of APC will not speed up but will be slower. As shown in Figure 11f, when $batch <= 2^{11}$, $C_a$ is small, and the acceleration benefit cannot exceed the compression overhead. When $batch > 2^{11}$, there is positive feedback.

Table 4: The impact of network communication on throughput.

| Env-2(4:16) | ResNeXt500 |
| --- | --- |
| WPipe-R-APC | 732.6/3328 |
| WPipe-R-NonAPC | 607.8/3328 |

For convolutional networks, $C_a$ is large, and APC has always had a positive effect, but the effect is only obvious when the batch size is large enough, as shown in Figure 11e. In summary, when

Table 5: The accuracy difference between using APC and not using APC with the same hyperparameters. We set $\nu = 8 \times 10^{-5}$, $b = 32 \times 8$, $ws = 200$, $epoch = 1$ for BERT$_{\text{BASE}}$ and $\nu = 4 \times 10^{-5}$, $b = 16 \times 8$, $ws = 100$, $epoch = 1$ for BERT$_{\text{LARGE}}$ and $\nu = 0.01$, $b = 32 \times 8$, $wr = 0.05$ for ResNeXt50_32x4d and ResNeXt101_32x8d.

|  | BERT$_{\text{BASE}}$ | BERT$_{\text{LARGE}}$ |
|---|---|---|
| $\triangledown$ F1 | -0.0003 ± 0.0011 | 0.0035 ± 0.0127 |
|  | RESNEXT50 | RESNEXT101 |
| $\triangledown$ ACC | -0.0008±0.0014 | 0.0012± 0.0012 |

the batch is large, we use the communication compression technique to have a positive benefit, but in many cases, we do not need to use such a large batch size to train the model, so in most cases, model-parallel communication will not be a bottleneck.

**Overlap of Computation and Communication.** As shown in Figure 13, the activations or gradients communication can overlap with the forward pass, backward pass, and activation recomputation. Especially for activation recomputation, its time overhead can be offset mostly.

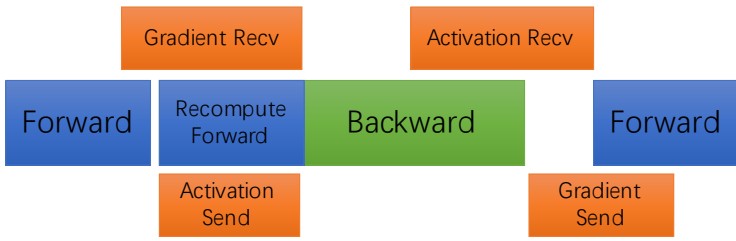

Figure 13: The overlap of model execution and activations/gradients.

**Heterogeneous Network Communication.** Generally, the intra-machine bandwidth (NVLink) is higher than the inter-machine bandwidth. In WPipe, the communication tasks mainly include model-parallel communication and data-parallel communication. Thus, WPipe allows tasks with large communication overhead to use the higher intra-machine bandwidth, and tasks with small communication overhead to use inter-machine bandwidth, to balance communication overhead.

## A.4 MODEL PARTITIONS GROUPING

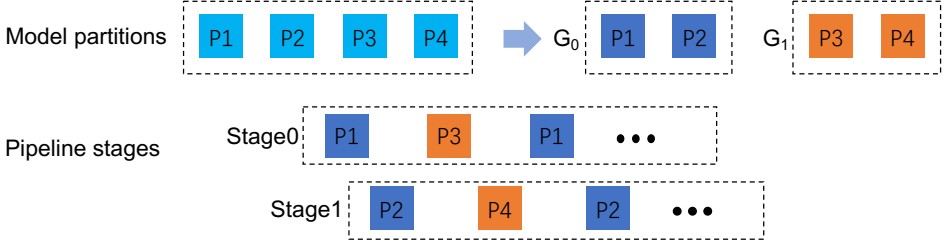

Figure 14: The relationship between model partitions and pipeline stages.

## A.5 EXPANSION

**GPipe Grouping.** As shown in Figure 15, the pipeline grouping technique can also be applied to GPipe, thereby reducing GPipe's bubbles. In Figure 15, the two model partitions are further divided

into four and divided into two groups. At this time, the bubble is reduced by half. For further discussion, when the number of groups is $N$, the bubble will be reduced to $\frac{1}{N}$, but the model-parallel communication overhead will increase by $N$ times. However, if NVLink can be used, the impact of increased communication overhead will be greatly reduced.

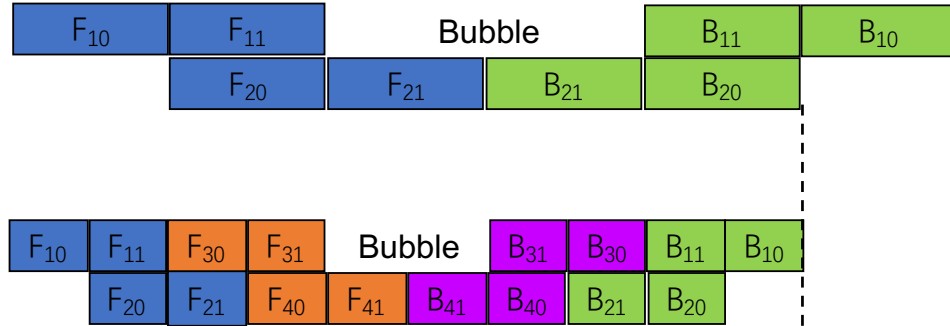

Figure 15: The pipeline grouping is applied to GPipe.

