# OpenReview forum: "Group-based Interleaved Pipeline Parallelism for Large-scale DNN Training"
_ICLR.cc/2022/Conference — ICLR 2022 Poster_

### Official Review · Reviewer_5TDo · 2021-11-02

**Correctness:** 4
**Technical Novelty And Significance:** 2
**Empirical Novelty And Significance:** 3
**Recommendation:** 8
**Confidence:** 4

**Main Review:**

Training very large DNNs models is a hot topic of practical interest. Data parallelism cannot help when the models are so big that they cannot fit in a single accelerator. In this case, model-parallel techniques can be applied. There are several existing pipelines, such as GPipe, PipeDream, PipeDream-2BW that improve various aspects of the naive execution. However, they typically suffer from (1) staleness of the weight updates, caused by inconsistencies in the versions of the weights used in the forward and backward pass, and (2) large number of in-flight activations, caused by the forward passes waiting for their backward passes in the pipeline.

The paper proposes a new pipeline called WPipe, that builds on the existing ones, but addresses the previously mentioned two issues. A careful construction is proposed, based on two groups of weights. A key feature of the new scheme is the placement of the forward pass of the next step for one of the groups (G0) at the front of the backward pass of G0 of the current step. This elegant construction, allows group G1 to only maintain one weight version and G0 to maintain only two weight versions, thus significantly reducing the memory overhead.

A theoretical analysis of the memory footprint is provided. WPipe can also benefit from activation recomputation techniques, that can further reduce the in-flight activations at the expense of more computation. Furthermore, the communication overhead is analyzed, and some mitigation strategies are suggested: data parallelism, overlap of computation and communication, heterogeneous network communication.

A significant part of the paper is dedicated to experimental analysis, including text classification (BERT models) and image classification (ResNeXt models). The experiments show that compared to state-of-the-art PipeDream-2BW, WPipe is more memory efficient (by36%) and has higher throughput (1.4x). Although the final accuracy is similar for the two schemes, WPipe is shown to have weight update semantics closer to data parallelism.

**Summary Of The Paper:**

The paper addresses the problem of very large scale deep neural network training through model parallelism. A new model parallel pipeline called WPipe is proposed, that builds on previously existing schemes while targeting (1) lower memory redundancy and (2) fresher weight updates. Experimental evaluation shows that WPipe can achieve significant acceleration (1.4x) and lower memory footprint (36% smaller) when compared to state-of-the-art PipeDream-2BW.

**Summary Of The Review:**

The paper proposes a new model-parallel pipeline called WPipe, for DNN training. A careful construction is proposed, based on two groups of weights. A key feature of the new scheme is the placement of the forward pass of the next step for one of the groups (G0) at the front of the backward pass of G0 of the current step. This elegant construction, allows group G1 to only maintain one weight version and G0 to maintain only two weight versions, thus significantly reducing the memory overhead. Theoretical analysis and an extensive experimental evaluation on text and image data show that WPipe outperforms state-of-the-art schemes, being 1.4 times faster while using 36% less memory.

---

> ### Author Response · Authors · 2021-11-20
> **Author Response**
>
> Thank you for your excellent review!

---

### Official Review · Reviewer_rrtJ · 2021-11-04

**Correctness:** 3
**Technical Novelty And Significance:** 4
**Empirical Novelty And Significance:** 2
**Recommendation:** 8
**Confidence:** 4

**Main Review:**

Strengths:
1. The proposed pipeline strategy is novel and effective.
2. Clean analysis of the proposed strategy in terms of memory footprint, communication overhead, bubble time, and update staleness.

Weakness:
There is no strong weakness of this paper. For other minor clarification and writing issues, see the comments below.

Detailed comments:
1. The term "model parallelism" should be used more precisely. There are three types of model parallelism: pipeline parallelism, operator placement (which is referred by you as "conventional model parallelism" or "naive model parallelism"), and operator partition (e.g., the strategy in Megatron-LM v1). In your paper, it seems you totally ignore the third type operator partition. I suggest renaming all "naive model parallelism", "conventional model parallelism" in your paper to "operator placement", and adding some description of the operator partition.
2. The memory footprint analysis with activation recomputation seems wrong. I think all numbers in the last column of Table. 1 are wrong. Take GPipe for example, activation recomputation only reduces the memory footprint of a micro-batch on a stage by a constant factor. It cannot reduce the footprint by a factor of "M", because anyway, each stage has to keep the input activation of the "M" in-flight micro-batches. The same analysis applies to other pipeline schemes.
3. What will happen if we split the weights into more groups instead of just 2 groups?
4. Please provide the exact settings of the models and the cluster. For example, provide the hidden_size, seq_len, num_heads of all BERT models. Provide the network bandwidth of your private cluster.
5. How do you split the models into pipeline stages? For BERT, you can assign an equal number of transformer layers to each stage. How do you split the ResNext?


**Summary Of The Paper:**

This paper introduces a novel pipeline training strategy WPipe. WPipe divides model partitions into two groups and updates each group alternatively, which eliminates half of the delayed gradients and memory redundancy compared to Pipedream-2BW.

The experimental results show that WPipe can achieve higher throughputs and reduce memory footprint with similar final model accuracy.

**Summary Of The Review:**

This paper introduces a novel pipeline training strategy and comprehensively evaluates its performance.
Although the writing can be improved, I recommend accepting this paper.

---

> ### Author Response · Authors · 2021-11-20
> **Explain "more groups", "memory footprint analysis with activation recomputation", and division strategy of the ResNeXt. Supplement intra-layer model parallelism and model settings.**
>
> Thank you for your excellent review!
>
> **Q1**. I suggest renaming all "naive model parallelism", "conventional model parallelism" in your paper to "operator placement", and adding some description of the operator partition.
>
> **A1**. Good suggestion. However, the "operator placement" and "operator partition" in our language environment may be a bit vague to readers, so we use inter-layer model parallelism (operator placement) and intra-layer model parallelism (operator partition) instead of them in the revision. In addition, we also supplemented the relevant background of intra-layer model parallelism in the Model Parallelism section of the BACKGROUND.
>
> **Q2**. The memory footprint analysis with activation recomputation seems wrong.
>
> **A2**. You are right, and it should contain input activations, but in general, their memory footprint is small, and we ignored them to simplify the analysis. As the title of Table 1 says "where we ignore the intermediate activations for recomputation."
>
> **Q3**. What will happen if we split the weights into more groups instead of just 2 groups?
>
> **A3**. Good question. We had intended to discuss this issue in-depth in the paper, but limited space. Here we provide a simple explanation: the number of groups is used to trade off the degree of reducing in-flight activations and weight update delay. If we use N to represent the number of groups (N>=2), the reduced in-flight activation is (n-1)/n, and the reduced update delay is 1/n. When N=2, the update delay is reduced the most.
>
> **Q4**. About settings of the models and the cluster.
>
> **A4**. The configuration of the model: hidden_size=768, seq_len=128, num_attention_heads=12 for all BERT models. Cluster bandwidth: the inter-machine bandwidth is 25Gbps, and the NVLink bandwidth is 300GBps. We have added them to the appendix of the revision. In addition, you can quickly reproduce experimental results through experimental code, runtime docker image, and operating instructions provided in the supplementary materials.
>
> **Q5**. How do you split the models into pipeline stages? How do you split the ResNeXt?
>
> **A5**. Compared to Bert, ResNeXt is also a multi-layer structure, and its difference is that its layers are divided into four blocks. We split each block by layer, and if the number of layers in the current block is insufficient, we will use the layers in the next block to supplement, and finally achieve to split the ResNeXt. For the detailed implementation, please refer to the code wpipe/cv/resnext_split.py provided in the supplementary materials. Of course, this simple division strategy will not achieve perfectly even division, which explains why the memory efficiency in our experiment is less than the theoretical memory efficiency.

---

### Official Review · Reviewer_fotX · 2021-11-06

**Correctness:** 4
**Technical Novelty And Significance:** 2
**Empirical Novelty And Significance:** 2
**Recommendation:** 6
**Confidence:** 4

**Main Review:**

There is some novelty in the new pipeline scheme, but I feel it is incremental. On the practical side, the method has the potential to be more impactful to large scale DNN transformer models such as BERT, GPT etc. The empirical results are very encouraging to me. The presentation and clarity in the paper could be improved - the authors should re-use terminology used earlier in the pipeline parallelism literature to make the presentation consistent and less confusing.

Pros:
- Simple idea that seems to work very well empirically. Can be readily adapted and used in existing training pipelines.
- Memory Footprint Analysis is provided for the method
- Empirical results are very encouraging

Cons:
- Novelty factor is bit limited
- Presentation could be improved
- Reproducibility can be improved (it would be benefit readers if the authors could provide hyper-parameters and experimental setup in a Table summarized for all of their runs)

Comments:
- In the abstract, authors mention - "It moves the forward pass of the next period ..." -> period is not defined at this point and is too vague a term to use without defining what it exactly refers to.

- "model partitions" -> do the authors mean pipeline stages? Strongly recommend using standard terminologies from model parallelism literature and explain things more consistently. I feel the paper is written a little hastily.

- ".. raw pipeline parallelism ... suffers from inconsistency issue" -> what does inconsistency mean here? It still is a bit vague term to me.

- How does WPipe react to varying the varying number of pipeline stages? What is the maximum number of pipeline stages the authors tested the method with? It would be useful to understand this scalability.





**Summary Of The Paper:**

The paper presents a novel pipeline parallelism technique for training large DNN models named as WPipe. The method aims to improve upon existing pipeline methods such as Pipedream-2BW by having a better memory efficiency and more fresh weight updates. Key idea is to divide the model partitions into two parts groups and make the forward pass of the second part to reduce the delayed gradients. The method is tested empirically and performs better than proposed current state of the art baselines.

**Summary Of The Review:**

The novelty factor in the proposed work is incremental and I also feel the paper can benefit from some more polishing of the draft due to the presentation issues I outlined.

---

> ### Author Response · Authors · 2021-11-20
> **Author Response**
>
> Thank you for your constructive and detailed review!
>
> **Q1**. The re-use terminology used earlier in the pipeline parallelism literature should make the presentation consistent. Presentation could be improved.
>
> **A1**. Good suggestion. In the revision, we have modified the batch in the gradient accumulation period to “micro-batch". And in the hybrid parallelism subsection, we used the description method of depth and width, which is consistent with the previous literature.
>
> **Q2**. About Reproducibility.
>
> **A2**. We have added the detailed experimental settings to the appendix of the revision. In addition, you can quickly build an experimental environment and reproduce our experimental results through the provided code, runtime docker image, and operating instructions provided in the supplementary materials.
>
> **Q3**. The period is not defined at this point and is too vague a term to use without defining what it exactly refers to.
>
> **A3**. Good suggestion. We have modified the "period" to the "period of weight updates" to avoid confusion.
>
> **Q4**. "model partitions" -> do the authors mean pipeline stages?
>
> **A4**. They are different. Regarding these two phrases, we follow the treatment in PipeDream and PipeDream-2BW. Specifically, they are inclusion relationships. In this paper, a pipeline stage contains two model partitions. In addition, in the memory footprint analysis subsection, we used "model partitions" incorrectly. Now we have modified them to "pipeline stages" in the revision.
>
> **Q5**. ".. raw pipeline parallelism ... suffers from inconsistency issue" -> what does inconsistency mean here? It still is a bit vague term to me.
>
> **A5**. We have modified the "inconsistency issue" to the "inconsistency issue discussed above" in the revision to better relate the above contents. Precisely, they correspond to lines 9-10 of the second paragraph of the INTRODUCTION above. And an example is also given in lines 3-6 in the Pipeline Parallelism section of the BACKGROUND above.
>
> **Q6**. How does WPipe react to varying the varying number of pipeline stages?
>
> **A6**. It depends on the relationship between model-parallel and data-parallel communication overhead. Specifically, when the number of GPUs remains the same, for models such as convolutional networks, whose model-parallel communication overhead is greater than that in data-parallel, WPipe will reduce the number of pipeline stages and expand the width of data parallelism. For other models such as the fully connected networks and Transformers, whose model-parallel communication overhead is less than that in data-parallel, WPipe will increase the number of pipeline stages and compress the width of the data parallelism.
>
> **Q7**. What is the maximum number of pipeline stages the authors tested the method with?
>
> **A7**. The maximum number of pipeline stages is the same as the number of GPUs. We have 64 GPUs on Env-2, and we tested WPipe with a maximum of 64 pipeline stages. In theory, for pipeline parallelism, the more pipeline stages, the more in-flight activations accumulate. On this issue, WPipe has an obvious advantage in that it has fewer in-flight activations when recomputing.

---

> > ### Comment · Reviewer_fotX · 2021-11-20
> > **Model Parallelism terminology**
> >
> > Thanks to the authors for their responses. It does clarify part of my questions.
> >
> > To build up on reviewers response A1, Reading other comments here, I do feel like due to the nature of this paper and available previous literature that is being referenced, some terminology cleanup in the draft will help make it more clearer and easy to relate to existing work. I would also like to throw a suggestion  here of introducing the parallelism types citing the following examples from literature.
> >
> > Intra-layer parallelism -> example is "Tensor model parallelism" (Megatron style tensor slicing or operator partitioning as it is also called),
> > Inter-layer parallelism -> most common example prevalent in literature for this seems to be "Pipeline model parallelism".
> >
> > I agree sticking with more general terminology (intra-layer and inter-layer) is better but needs to be mapped to existing strategies as well.
> >
> > Regarding A4, I think the authors should clarify this with a diagram or rewrite to make it more clear. I am still bit confused by how the proposed method uses pipeline stages and model partitions.

---

> > > ### Author Response · Authors · 2021-11-21
> > > **Explain that pipeline parallelism is based on the improvement of inter-layer model parallelism, and further explain model partitions and pipeline stages.**
> > >
> > > Thank you for your insightful review again!  The latest revisions are marked in red. You can check it. Below are the specific response to your questions.
> > >
> > > **Q1**. I agree sticking with more general terminology (intra-layer and inter-layer) is better but needs to be mapped to existing strategies as well.
> > >
> > > **A1**. Good suggestion. We expand the "conventional model parallelism" in the initial version into "intra-layer model parallelism" and "inter-layer model parallelism", while the inter-layer model parallelism does not represent the pipeline parallelism, and the pipeline model parallelism is an improvement based on the inter-layer model parallelism. We have made these correspondences clearer in the latest revision.
> > >
> > > **Q2**. Regarding A4, I think the authors should clarify this with a diagram or rewrite it to make it more clear. I am still bit confused by how the proposed method uses pipeline stages and model partitions.
> > >
> > > **A2**. Good suggestion. We have added Figure 14 (at page 14 ) to illustrate the relationship between model partitions and pipeline stages. Due to space limitations, we temporarily put it in the appendix of the latest revision. In fact, Figure 5 in the paper already reflects the relationship between pipeline stages and model partitions to some extent. As shown in Figure 5, each row represents a pipeline stage, and each column represents the execution cell of a model partition at a certain time. In the latest revision, we have also added a more detailed description. Specifically, the pipeline stage represents the running stage of the pipeline. Generally, a model partition is executed in a pipeline stage, but in WPipe, two model partitions (included in G0 and G1, respectively) are executed alternately in a pipeline stage. Therefore, in other pipeline-parallel methods, their relationship is $N_{pipeline-stages} = N_{model-partitions}$, and in WPipe it is $N_{model-partitions}=2N_{pipeline-stages}$

---

### Official Review · Reviewer_VWfQ · 2021-11-08

**Correctness:** 3
**Technical Novelty And Significance:** 2
**Empirical Novelty And Significance:** 2
**Recommendation:** 3
**Confidence:** 3

**Main Review:**

The paper has following strengths.
1. It tackles an important problem of how to fit rapidly growing model sizes into slower growing device memory. Pipeline parallelism has been a useful approach for this problem, and this paper provides some improvements in that space.
2. It proposes an interesting approach to address key issues surrounding memory efficiency and weight update sematics in existing pipeline parallelism approach

The weakness include:
1. The overall appears narrow and incremental in the sense that WPipe is fixing a memory efficiency bug of PipeDream-2BW relative to PipeDream-flush. In other words, WPipe is trying to deliver the best-of-both-worlds for these two existing approaches. As a result, it is not obvious how the results applicable beyond the PipeDream* line of pipeline parallelism.

2. The throughput results (4.2) are difficult to understand because they are based on unfamiliar models (e.g., Bert768 and ResNetXt500) which are not previously defined and different from the familiar models (e.g., Bert_base, ResNetXt50) used for the convergence quality results (4.1). It seems that the models in 4.2 were constructed to show the best case throughput benefits of WPipe, but don't appear to be real world models.

3.  I feel the writing could be greatly improved as I found multiple portions difficult to follow. Although I am not hands-on with pipeline parallelism, I feel I have a pretty solid understanding at the high-level, but yet this draft was less accessible than expected. Below are some specific examples of confusing portions and questions for the authors:

  a) The idea (stated in 3.2) of splitting the model partitions to double the number of partitions and obtain two groups (G0 & G1) is quite confusing, especially since memory footprint analysis in 3.3 assumes the model is evenly divided so that G0 and G1 are half of the model.  It would be helpful to illustrate the before/after of model partitions with WPipe relative to PipeDream-2BW, independent of mini- or micro-batches. This should be possible since model partitions are simply mappings of model parameters to devices.

  b)  It seems the cells in Figures 1 & 2 are mini-batches and micro-batches respectively. It would be helpful to clarify this detail since these two Figures are often discussed together. Also, it seems Figure 2(c) is missing the update cell. Is this a typo?

  c) Much of the material in 3.4 is prior work and so could be written more briefly and systematically.

**Summary Of The Paper:**

The paper proposes, WPipe, a technique for reducing the memory overheads of weight versioning and improving freshness of weight updates in pipeline parallelism through a novel scheduling of the the pipeline stages. It presents evaluation results which show that WPipe does not harm model quality, and that the memory efficiency can improve the throughput compared to prior pipeline approaches.

**Summary Of The Review:**

WPipe is basically a memory efficiency optimization of PipeDream-2BW, but the paper does not report throughput benefits on real world models. So it is not clear how practical the optimizations are.

---

> ### Author Response · Authors · 2021-11-19
> **Author Response**
>
> Thank you for your insightful review!
>
> **Q1**. The overall appears narrow and incremental in the sense that WPipe is fixing a memory efficiency bug of PipeDream-2BW relative to PipeDream-flush.
>
> **A1**. The memory efficiency of PipeDream-2BW is not a bug relative to PipeDream-flush. In fact, it is the result of the trade-off between memory footprint and throughput made by PipeDream-2BW. Although PipeDream-2BW needs to maintain one more weight version, it eliminates the bubble in the pipelining of PipeDream-flush and achieves higher throughput.
>
> WPipe makes one step further. Via a carefully designed group-based pipelining parallelism, it further reduces the memory redundancy of PipeDream-2BW by half, without introducing any extra bubbles needed in PipeDream-flush. As a result, it enjoys the best of both PipeDream-2BW and PipeDream-flush. Its construction is not a simple mixing of both. Note that the novelty and significance of WPipe have also been well approved by most other reviewers.
>
> **Q2**. The throughput results are based on unfamiliar models, but convergence quality results are based on familiar models. It seems that the models in 4.2 were constructed to show the best case throughput benefits of WPipe, but don't appear to be real-world models.
>
> **A2**. Our experimental design largely follows that of PipeDream-2BW. In this design, the small models (or familiar models) are merely used to evaluate the convergence of WPipe. This is because WPipe needs to be compared to the data parallelism, with which larger models cannot be trained. In contrast, the throughput experiments require larger models to better illustrate the performance of the training system. After all, WPipe is designed to train large models.
>
> The unfamiliar models Bert768 and ResNeXt500 are obtained by increasing the number of layers of Bert_base and ResNeXt50. Except for the number of layers, all other configurations remain unchanged (hidden_size=768, seq_len=128, num_attention_heads=12 for Bert, and groups=32, width_per_group=4 for ResNeXt). This adjustment is just to increase the parameter scale of the model. In addition, we have done experiments on models with varying layer numbers. They are Bert96, Bert192, Bert384, Bert768, ResNeXt200, ResNeXt302, ResNeXt500, and ResNeXt800, which can reflect the universality of the experimental results.
>
> **Q3**. The idea (stated in 3.2) of splitting the model partitions to double the number of partitions and obtaining two groups (G0 & G1) is quite confusing, especially since memory footprint analysis in 3.3 assumes the model is evenly divided so that G0 and G1 are half of the model. This should be possible since model partitions are simply mappings of model parameters to devices.
>
> **A3**. In general, G0 and G1 are not just simply mapped to the corresponding devices. Actually, each of them has its own specific execution logic, and they can be regarded as relatively independent sub-pipelines.
>
> Specifically, G0 and G1 run alternately, and each controls its own update logic. G0 needs to periodically switch between two weight versions and use the gradient of the previous period to update the current weight version. In contrast, G1 does not need to switch between different weight versions because it only maintains a single weight version. Hence it uses the gradient of the current period to update the weight version.
>
> In addition, for G0, the execution order of its forward and backward passes is also adjusted. Namely, the forward pass of the next gradient accumulation period will be moved to the front of the backward pass of the current gradient accumulation period.
>
> Regarding the assumption that G0 and G1 are evenly divided, it is a simplified assumption to aid the analysis of the upper limit of the memory redundancy reduction rate. The theoretical results show that, at best, WPipe can reduce the memory redundancy by 50% compared to PipeDream-2BW. In experiments, since G0 and G1 are not perfectly evenly divided, WPipe reduces the memory redundancy by 36%.
>
> **Q4-1**. It seems the cells in Figures 1 & 2 are mini-batches and micro-batches respectively.
>
> **A4-1**. Good suggestion. For the representation of mini-batches, we follow the treatment in PipeDream. But PipeDream does not use gradient accumulation. When using gradient accumulation, it is better to use micro-batches to represent the batch of each execution cell in the accumulation period. We have modified the relevant contents in the revision.
>
> **Q4-2**. Also, it seems Figure 2(c) is missing the update cell. Is this a typo?
>
> **A4-2**. Figure 2(c) shows the pipelining of PipeDream. Each backward pass of PipeDream will update the parameters, so we did not indicate the update cell.
>
> **Q5**. Much of the material in 3.4 is prior work and so could be written more briefly and systematically.
>
> **A5**. Good suggestion. In the revision, we have compressed 3.4 and put the detailed description in the appendix.

---

> > ### Comment · Reviewer_VWfQ · 2021-11-22
> > **Thanks, but concerns remain**
> >
> > Thanks for your response.
> >
> > **A1** : This response provided useful clarification.  However, it also reinforces my concern that these results are narrow in scope and application as it confirms that WPipe is a best-of-both-worlds for PipeDream-2BW and PipeDream-Flush (sentiments expressed earlier in **Q1**). I apologize that my phrasing, "memory efficiency bug of PipeDream-2BW relative to PipeDream-Flush" was imprecise and thus confusing, but it seems you got the main point. Which is that the memory cost in PipeDream-2BW of eliminating PipeDream-Flush bubbles was unnecessarily high. WPipe seems to confirm this point by showing that 50% of that memory cost is theoretically avoidable.
> >
> > **A2**: The concern here is that increasing only the layer count of a Bert model does not reflect scaling of NLP models in practice, where other architecture parameters like dimensionality and attention heads are also scaled. It seems more realistic throughput results could have been collected using SOTA NLP models like GPT and T5, which are larger than Bert, and whose multi-billion parameter versions are publicly available.

---

> > > ### Author Response · Authors · 2021-11-22
> > > **Explain some possible misunderstandings.**
> > >
> > > Thank you for expressing your concerns in more detail, so that we can understand your thoughts better. You may have some misunderstandings about WPipe.
> > >
> > > Regarding **A1**, WPipe, like GPipe, PipeDream-2BW, PipeDream-flush, etc., is a general pipeline parallel training system. Where the others can be used, WPipe can also be used. In addition, regarding the pipeline grouping method, it is novel and widely applicable. We can even use it to improve GPipe and reduce its bubble by at least half. For details, we have added part A.4 in the appendix of the latest revision and marked it in blue.
> > >
> > > Regarding **A2**, as a general pipelined parallel training system, WPipe does not care what the trained model is, it only needs to care about the number of model parameters and intermediate activations because they determine the data-parallel communication overhead and model-parallel communication overhead, respectively, which affects the final throughput. Compared to other methods, WPipe is more sensitive to intermediate activations, so we use the Transformer model with small intermediate activation and the ResNeXt model with large intermediate activation to test separately. In fact, Bert, GPT, and T5 are essentially the same type and are based on the Transformer structure. Their intermediate activation is generally small, and the parameter scale is generally large. As a result, their test results should be similar. ResNeXt's intermediate activation is larger, and the parameter scale is smaller, which can better explain the influence of model-parallel communication on WPipe.

---

### Decision · Program_Chairs · 2022-01-20

**Decision:**

Accept (Poster)

**Comment:**

The paper proposes a new pipeline-parallel training method called WPipe. WPipe works (on a very high level) by replacing the two-buffer structure of PipeDream-2BW with a two-partition-group structure, allowing resources to be shared in a similar way to PipeDream-2BW but with less memory use and less delays in weight update propagation across stages. The 1.4x speedup it achieves over PipeDream-2BW is impressive.

In discussion, the reviewers agreed that the problem WPipe tries to tackle is important and that the approach is novel and interesting. But there was significant disagreement among the reviewers as to score. A reviewer expressed concern about the work being incremental and difficult to follow. And while these were valid concerns, and the authors should take note of them when revising their paper, I do not think they should present a bar to publication, both based on my own read of the work and also in light of the fact that other reviewers with higher confidence scores did not find novelty to be a disqualifying concern. As a result, I plan to follow the majority reviewer opinion and recommend acceptance here.